# Efficacy of regular gargling with a cetylpyridinium chloride plus zinc containing mouthwash can reduce upper respiratory symptoms

**Francisco Wilker Mustafa Gomes Muniz**[1,2*], **Maísa Casarin**[1,2], **Natália Marcumini Pola**[1,2], **Cassiano Kuchenbecker Rösing**[3], **Taciane Menezes da Silveira**[2], **Francisco Hecktheuer Silva**[2], **Guilherme Azario de Holanda**[2], **Larissa Viana de Oliveira**[4], **Pedro Paulo de Almeida Dantas**[2], **Bernal Stewart**[5,6], **Zilson Malheiros**[5,6], **Carlos Benítez**[6], **Lyndsay Schaeffer**[5]

**1** Department of Periodontology, Federal University of Pelotas, Pelotas, Rio Grande do Sul, Brazil, **2** Graduate Program in Dentistry, Federal University of Pelotas, Pelotas, Rio Grande do Sul, Brazil, **3** Department of Periodontology, Federal University of Rio Grande do Sul, Porto Alegre, Rio Grande do Sul, Brazil, **4** School of Dentistry, Federal University of Pelotas, Pelotas, Rio Grande do Sul, Brazil, **5** Colgate-Palmolive Company, Piscataway, New Jersey, United States of America, **6** Latin American Oral Health Association, São Paulo, Brazil

* muniz.fwmg@ufpel.edu.br

## Abstract

The posterior oral cavity serves as an entry point to both the digestive and respiratory systems. Optimal oral hygiene, particularly by means of gargling, has been shown to effectively reduce transmission and duration of respiratory illnesses (Clinical Trials registration: NCT06479226). Previous studies have demonstrated the effectiveness of gargling with an antimicrobial mouthwash in preventing the development of respiratory symptoms. To assess the impact of using an antimicrobial mouthwash for gargling in lowering the incidence of upper respiratory symptoms. A total of 150 individuals were recruited and randomly assigned to two groups: Control group: participants were requested to brush their teeth twice daily for two minutes using a commercially available fluoride toothpaste; and the Test Group, which also brushed their teeth twice daily for two minutes with a commercially available fluoride toothpaste and additional gargling twice a day with a mouthwash containing 0.075% Cetylpyridinium chloride and 0.28% Zinc Lactate (CPC+Zn). Individuals were asked to complete the WURSS-21 Daily Symptom Report over the course of the three-month study period using a diary. Regular gargling with the mouthwash containing CPC+Zn was efficacious in decreasing both the frequency (21.5% lower) and severity (11% lower) of respiratory symptoms throughout the study. Additionally, individuals that performed consistent cleansing of the posterior oral cavity with mouthwash had greater interference in daily activities, which should be further investigated. Adding gargling with a mouthwash containing 0.075% CPC + 0.28% Zn to a normal oral hygiene routine proves beneficial in lowering the incidence of upper respiratory symptoms commonly associated with cold and the flu.

**Data availability statement:** All relevant data are within the manuscript and its Supporting information files.

**Funding:** This study was sponsored by Latin American Oral Health Association and Colgate-Palmolive. The study was also partially funded by Coordenação de Aperfeiçoamento de Pessoal de Nível Superior-Brasil, Finance Code - 001.

**Competing interests:** Drs. Stewart, Malheiros, and Schaeffer are employed by the Colgate-Palmolive Company. Dr. Benítez is employed by the Latin American Oral Health Association. Dr. Rösing and Dr. Muniz hold research scholarships from the National Council for Scientific and Technological Development – CNPq.

**Trial registration:** ClinicalTrials.gov NCT06479226

## Introduction

Respiratory infections, whether they are acute or chronic, occur frequently in both adults and children [1]. Globally, yearly epidemics of viral respiratory infections pose a significant health challenge, leading to increased economic strain on healthcare systems, heightened morbidity, and higher mortality rates [1,2]. In 2019, the number of upper respiratory infections reached 17.2 billion cases, accounting for 42.82% of all diseases and injuries included in the Global Burden of Diseases for that year [3]. The global prevalence of respiratory illnesses, including seasonal flu, stands at approximately 1 billion cases per year. Among these cases, approximately 3 to 5 million instances are classified as severe flu infections [4].

In the United States, the estimated economic costs linked to common cold amounted $40 billion annually, whereas influenza incurs an even higher economic burden, surpassing $87 billion each year [5,6]. A population-based cross-sectional study conducted in Brazil in May 2020 reported a prevalence of flu-like syndrome symptoms of 3.38% [7]. Furthermore, influenza incidence in Brazil is influenced by seasonal variation, with higher rates observed in winter and the lowest rates recorded in January during the summer months [8]. Regarding the factors associated with upper respiratory infections, literature indicates that the presence of high humidity and low temperatures is related with several virus-related infections worldwide [9]. Furthermore, the oral cavity serves as a gateway for bacteria and viruses, facilitating their entry into the lower airways, such as the bronchi and lungs, which can be affected by respiratory diseases [10]. Research has also demonstrated that maintaining good oral health care can be effective in preventing the onset of respiratory diseases [10].

Literature also reports virucidal efficacy of chlorhexidine (CHX), as this product has demonstrated the ability to reduce viral load [11]. It has also proven effective against viruses like Influenza A [11,12]. Moreover, the virucidal effectiveness of cetylpyridinium chloride (CPC) has been previously documented, which includes reducing the duration of flu symptoms such as cough and sore throat [13,14]. Additionally, it has been reported that the salivary SARS-CoV-2 viral load was significantly reduced after a single rinse with mouthwashes containing CHX or CPC with zinc (Zn) [15]. CPC+Zn are antiseptic components present in mouthwashes, which are safe to use and have demonstrated excellent antiplaque and antigingivitis efficacy as adjuvants to mechanical oral hygiene practices [16,17].

The present study assessed the efficacy of regular gargling with a mouthwash containing CPC+Zn in reducing the incidence of upper respiratory symptoms. The working hypothesis was that individuals who regularly cleansed the posterior oral cavity by gargling with mouthwash would experience a lower incidence of upper respiratory symptoms compared to those who did not.

## Materials and methods

This is a Phase III, open-label, randomized clinical trial that employed a single-blind, parallel design. The study protocol had previously received approval from the Ethics Committee of the School of Dentistry at the Federal University of Pelotas, under protocol CAAE 58924622.7.0000.5318 (S1 and S2 Appendixes). All participants read and signed an informed consent form before being enrolled in the study. Therefore, verbal and written consent was obtained. The informed consent was approved by the local Ethics Committee. The study protocol was registered *a posteriori* in Clinical Trials database (NCT06479226) (S3 Appendix). As

this is an innovative methodology for oral hygiene products, it was decided not to register its protocol prior to data collection for industrial purposes.

No deviation from protocol is applied to the current study. The authors confirm that all ongoing and related trials for this drug/intervention are registered. The raw data is available in the S4 Appendix. The study followed the CONSORT checklist (S5 Appendix).

## Participants, inclusion and exclusion criteria

The study was conducted between July 4th and October 6th 2022, and a consecutive sampling was recruited (Recruitment happened between July 4th to July 15th, while follow-up period finished in October 6th 2022). Participants were retrieved from the lists of patients of the School of Dentistry – Federal University of Pelotas, Pelotas, Brazil.

In order to be included, participants had to meet all of the following criteria:

1) Male or female aged 18 to 70 years of age;

2) Be in good general health as determined by the study investigators, which included the absence of any respiratory symptoms at baseline. Participants with upper respiratory symptoms were not involved, as the current study did not aim to treat these conditions;

3) Available for the duration of the study.

Participants who had any of the following conditions were excluded:

1) Participation in any other oral clinical study for the duration of this study;

2) Self-reported pregnancy and/or currently breastfeeding;

3) Allergies to oral care products, personal care consumer products, and/or their ingredients;

4) Currently experiencing oral irritation or using oral anesthetics;

5) Self-reported history of diabetes or use of any hypoglycemic drug;

6) Submitted to oral surgery or extensive dental work during this study;

7) Immune compromised (HIV, AIDS, immuno-suppressive drug therapy);

8) Wear full dentures;

9) Carpal tunnel or arthritis in their hands. These individuals were excluded as literature shows poorer oral hygiene behaviors, including higher prevalence of oral diseases, among those with arthritis [18]. Moreover, it was hypothesized that those with carpal tunnel syndrome would perform their oral hygiene in a lower efficiency when compared to those without it.

## Experimental groups

Two experimental groups were employed in this study. In the control group, participants were provided with a commercially available adult soft bristle toothbrush and a toothpaste containing fluoride. In the test group, participants received the same toothbrush and toothpaste. In addition, a mouthwash containing 0.075% CPC + 0.28% Zn lactate in an alcohol-free base was used. Patients were instructed to use only the provided products, though interproximal hygiene was permitted. No placebo substance was used in the control group, as the current study aimed to compare individuals who perform the cleansing of their throat with CPC+Zn in comparison to those who did not perform it.

In both groups, participants were given instructions to brush their teeth for two minutes, twice daily (morning and evening), using the provided toothpaste. Additionally, only in the test group, after brushing their teeth, participants were requested to gargle with the provided mouthwash. They were asked to pour 20 ml of the mouthwash, gargle for 30 seconds, and then spit it out. This procedure was to be repeated twice daily throughout the 90-day follow-up period.

The distribution of the products was performed in a distinct area from the examination room by site personnel who were not part of the clinical evaluations (MC, NMP, and TMS). Oral hygiene products were provided to participants in sealed bags to ensure consistency in product aesthetics and packaging across study groups. The instructions given to participants included a study group code, guidance for at-home use, and safety information, which encompassed emergency contact details. The treatment products were replenished 30 days after baseline. No dietary restrictions were imposed on participants during the study. Upon finishing the 90-day follow-up period, participants were instructed to return all both used and unused products.

### Clinical examination

During the initial baseline appointment, participants received an oral soft tissue assessment aimed at identifying any abnormal conditions in various areas, including the soft palate, hard palate, gingival mucosa, buccal mucosa, mucogingival folds, tongue, sublingual and submandibular salivary glands, tonsillar region, and pharyngeal areas. This examination was conducted by a trained and blinded examiner (FWMGM). The same clinical examination was repeated after 30 and 90 days of follow-up.

### Outcome

The primary outcome of this study was the incidence of upper respiratory symptoms linked to the cold and flu season. Participants were required to diligently fill out the WURSS-21 Daily Symptom Report [19] on a daily basis during the 3 months of follow-up. This questionnaire consists of 21 items designed to evaluate the overall severity of upper respiratory symptoms. It employs a Likert scale, with responses ranging from 0 (indicating no symptoms) to 7 (indicating the most severe symptoms).

The first 11 questions assess symptom severity, encompassing overall symptoms (with the following question: "How sick do you feel today?") and the severity of specific symptoms experienced in the last 24 hours. These symptoms include runny nose, congested nose, sneezing, sore throat, scratchy throat, cough, hoarseness, head pressure, chest pressure, and fatigue. Additionally, the questionnaire evaluates how these symptoms interfere with daily activities, which encompass the ability to think clearly, sleep well, breathe easily, engage in physical exercise, perform routine daily tasks, work outside the home, work within the home, interact with others, and maintain one's personal life.

To encourage compliance to the protocol, subjects were requested to submit photos of the questionnaire on a weekly basis. During these interactions, they were also reminded to consistently use the designated oral hygiene products allocated to them.

### Sample size

The study aimed to include a total of 150 participants, with the expectation that at least 136 (68 in each group) would successfully complete the study protocol. A power calculation indicated that with approximately 75 replicates for each participant, the study can detect a

difference in incidence rates of 3% with 80% probability. This calculation factored in an estimated dropout rate of approximately 10% over the course of the study.

## Randomization and allocation concealment

The randomization process was performed by a researcher (NMP) who was not part of the clinical examinations. Simple randomization was employed, and a website (https://www.randomization.org) was utilized for this purpose. This process was performed in blocks of different sizes. However, participants who cohabited were assigned to the same study group. Allocation concealment was also managed by the same researcher, who used opaque and sealed envelopes, which were stored in a room with limited access, to ensure the integrity of the allocation process.

## Statistical analysis

The individual was considered the unit of analysis. The raw symptom severity intensity score data were rescaled to a binary presence/absence score for statistical analysis. Therefore, scores of 1 or 0 were dichotomized in the data set. This strategy was performed to clearly demonstrate any impact of respiratory symptoms.

Interval plots with 95% confidence intervals were created to facilitate between treatment group symptom rate comparisons. T-test for independent sample was used to statistically compare the treatment group symptom rates. A Group p value less than 0.05 in the t-test indicated statistical significance.

## Results

Seventy-five individuals were enrolled in each experimental group. The mean ages were 31.63 ± 10.68 in the test group and 31.52 ± 9.81 in the control group. In total, 45 (60.00%) females were included in the test group, and 43 (57.33%) females in the control group. Four participants were lost to follow-up during the study, as demonstrated in Fig 1.

Four patients reported adverse events, with three of them occurring in the test group. None of these individuals were excluded from the study. In the control group, the adverse event was unrelated to the study protocol, as the patient required antibiotics due to a previously scheduled abdominal surgery. In the test group, the following adverse events were reported:

1) One patient in the test group experienced stomach pain. However, its cause was confirmed through an endoscopy examination, which revealed chronic gastritis resulting from *Helicobacter pylori* infection. The study protocol was not found to be causally related to this adverse event.

2) Another patient in the test group reported persistent tonsillitis, which was treated by a physician with antibiotics. The adverse event was probably not related to the protocol.

3) One patient reported a burning sensation after gargling with the mouthwash on the first day of product exposure. The individual continued to follow the protocol during the study, he was monitored. The adverse event was classified as low severity and considered to be probably related to the protocol.

Regarding overall respiratory symptoms, the mean scores were 0.29 ± 0.45 in the control group and 0.23 ± 0.42 in the test group ($p < 0.001$). This indicates that individuals in the test group, who included regular gargling with CPC+Zn in their oral hygiene routine, exhibited lower respiratory symptoms. Specifically, the test group experienced a 21.5% reduction in

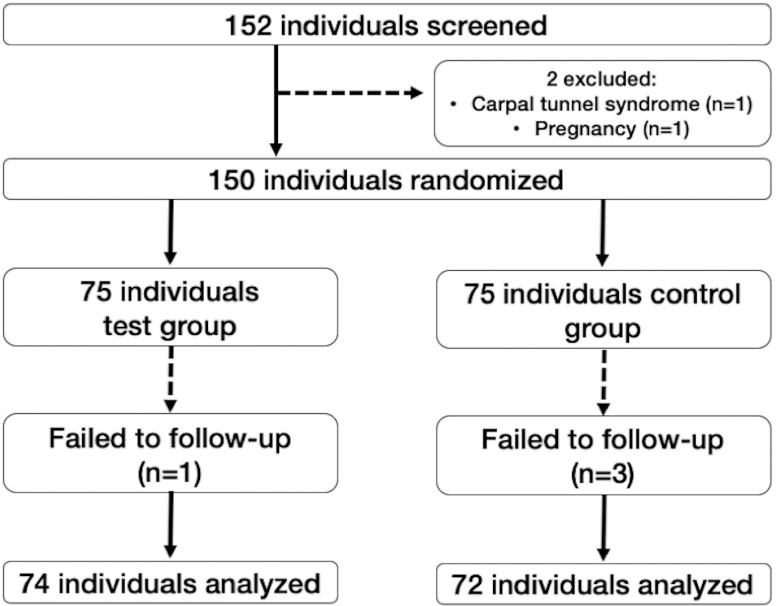

**Fig 1. Flowchart of participants during the study.**

overall respiratory symptoms and an 11% reduction in the severity of respiratory symptoms compared to the control group. When these symptoms were analyzed individually, the test group also demonstrated lower symptom severity in several of them, as detailed in Table 1. Individuals in the test group felt less sick during the study period and presented less runny nose, congested nose and cough. Hoarseness and chest pressure were less evident in the control group.

The test group experienced significantly higher self-reported impacts in daily activities caused by upper respiratory symptoms. This was particularly evident in questions related to thinking clearly, getting quality sleep, engaging in exercise, performing daily tasks, working

**Table 1. Mean±SD scores for each respiratory symptom according to the experimental group.**

|  | Test group | Control group | Difference | 95% CI (Test - control) | | P-value |
|---|---|---|---|---|---|---|
|  |  |  |  | Lower CI | Upper CI |  |
| How sick do you feel today? | 0.163 ± 0.369 | 0.184 ± 0.387 | −0.021 | −0.034 | −0.007 | **0.002** |
| Runny nose | 0.153 ± 0.360 | 0.178 ± 0.382 | −0.025 | −0.038 | −0.012 | **<0.001** |
| Congested nose | 0.127 ± 0.333 | 0.148 ± 0.355 | −0.021 | −0.033 | −0.009 | **0.001** |
| Sneeze | 0.096 ± 0.294 | 0.097 ± 0.296 | −0.001 | −0.012 | 0.009 | 0.808 |
| Sore throat | 0.065 ± 0.247 | 0.071 ± 0.256 | −0.006 | −0.014 | 0.003 | 0.224 |
| Scratch throat | 0.076 ± 0.265 | 0.079 ± 0.269 | −0.003 | −0.012 | 0.007 | 0.598 |
| Cough | 0.095 ± 0.293 | 0.107 ± 0.309 | −0.012 | −0.023 | −0.002 | **0.023** |
| Hoarseness | 0.061 ± 0.239 | 0.041 ± 0.199 | 0.020 | 0.012 | 0.027 | **<0.001** |
| Head pressure | 0.048 ± 0.214 | 0.046 ± 0.209 | 0.002 | −0.005 | 0.009 | 0.588 |
| Chest pressure | 0.013 ± 0.112 | 0.008 ± 0.089 | 0.005 | 0.001 | 0.008 | **0.010** |
| Tiredness | 0.076 ± 0.266 | 0.073 ± 0.259 | 0.004 | −0.005 | 0.013 | 0.413 |

**Legend:** Bold p-values mean statistically significant differences between groups (p < 0.05).

**Table 2.** Mean±SD scores for interference daily according to the experimental group.

| | Test group | Control group | Difference | 95% CI (Test - control) | | P-value |
|---|---|---|---|---|---|---|
| | | | | Lower CI | Upper CI | |
| Think clearly | 0.212 ± 0.409 | 0.154 ± 0.361 | 0.059 | 0.032 | 0.086 | **<0.001** |
| Sleep well | 0.485 ± 0.500 | 0.437 ± 0.496 | 0.048 | 0.012 | 0.083 | **0.008** |
| Breath easily | 0.428 ± 0.495 | 0.400 ± 0.490 | 0.029 | −0.006 | 0.064 | 0.101 |
| Exercise | 0.263 ± 0.440 | 0.162 ± 0.369 | 0.101 | 0.072 | 0.129 | **<0.001** |
| Perform daily activities | 0.272 ± 0.445 | 0.179 ± 0.383 | 0.093 | 0.064 | 0.122 | **<0.001** |
| Work outside | 0.233 ± 0.423 | 0.133 ± 0.340 | 0.100 | 0.074 | 0.127 | **<0.001** |
| Work inside | 0.228 ± 0.419 | 0.119 ± 0.324 | 0.108 | 0.082 | 0.134 | **<0.001** |
| Interact | 0.202 ± 0.402 | 0.164 ± 0.370 | 0.038 | 0.011 | 0.065 | **0.006** |
| Live life | 0.204 ± 0.403 | 0.134 ± 0.341 | 0.070 | 0.044 | 0.096 | **<0.001** |

**Legend:** Bold p-values mean statistically significant differences between groups (p < 0.05).

outside the home, working within the home, interacting with others, and maintaining a personal life, as indicated in Table 2.

## Discussion

The primary objective of this study was to evaluate the impact of regular gargling with an antimicrobial mouthwash on the reduction of upper respiratory symptoms. The findings consistently revealed that performing an additional step in oral hygiene by gargling with 0.075% CPC and 0.28% Zn Lactate significantly decreased the occurrence and severity of several respiratory symptoms. Based on feedback from test subjects that they experienced higher impact caused by upper respiratory symptoms in their daily activities, further investigation with a larger population is warranted. It is important to emphasize that this study marks one of the first explorations of the effects of gargling with an antimicrobial mouthwash on upper respiratory symptoms, making its findings particularly noteworthy.

The oral biofilm serves as a reservoir for respiratory pathogens, which are subsequently linked to other diseases, such as pneumonia. In this context, a systematic review demonstrated that maintaining good oral hygiene and receiving regular professional oral health care can reduce the progression and occurrence of respiratory diseases among high-risk older adults and individuals in intensive care units [20]. In fact, maintaining oral hygiene has been linked to prevention of influenza infection [21,22].

Previous studies explored CPC and its effectiveness, as it exhibits a high antibacterial potential and has demonstrated the absence of serious adverse effects [23,24]. A systematic review revealed a significantly higher antiplaque and antigingivitis efficacy of this compound when used as a supplement to mechanical oral hygiene [23,25]. The clinical superiority of CPC+Zn, in comparison to CPC only, is also demonstrated in the literature [17]. In addition, one study showed that CPC+Zn presented higher antiplaque and antigingivitis efficacy when compared to essential oils [16].

The use of a chemical solution in patients with chronic obstructive pulmonary disease resulted in an improvement in respiratory health-related quality of life when compared to the placebo group [26]. Moreover, performing oral hygiene that involves both antiseptics and toothbrushing has proven to be effective in reducing the incidence of ventilator-associated pneumonia and shortening the length of stay in intensive care units [27]. Hence, it was hypothesized that employing an antiseptic such as CPC+Zn, whether as a rinse or for gargling,

may potentially lower the bacterial or virucidal load or delay its subsequent increase. This, in turn, could potentially have a positive impact on respiratory symptoms.

It is crucial to take into account that the seasonality of respiratory infections tends to be more prominent during periods of low temperatures [28]. Recognizing how climatic fluctuations can directly impact the variables of the current study, all data collection was conducted within a narrow timeframe, specifically during the winter in the southern hemisphere. This ensured that all individuals were exposed to the same climatic conditions. Additionally, it is worth noting that Pelotas, Brazil, where the study took place, encounters high levels of relative humidity. This factor may have contributed to an environment conducive to the development of respiratory infections in this population [29].

The study's findings, demonstrating an overall reduction of 21.5% in respiratory symptoms and an 11% decrease in their severity in the test group compared to the control group, represent a significant outcome that underscores the importance of additional cleansing that targets the posterior oral cavity. Moreover, several specific symptoms, including "how sick do you feel today?," "runny nose," "congested nose," and "cough" exhibited statistically significant differences in favor of the test group. These symptoms can lead to financial costs and exert a significant impact on an individual's overall quality of life [30]. Literature has already reported that flu symptoms can impact daily activities [31,32]. Given that this additional cleaning through mouthwash is a straightforward and convenient technique, it could be considered as a recommendation for preventing upper respiratory infections. However, higher perception of symptoms was perceived in the test group regarding chest pressure and hoarseness. It may be suggested that patients in this group may have higher expectations of improvement due to the treatment, which might be attributed to novelty of the intervention.

The data indicated that the Control group generally exhibited more frequent cold/flu symptoms. This is a typical outcome one would expect from administering an efficacious preventative treatment. Interestingly, when afflicted individuals from both groups were asked about the degree to which symptoms interfered in their day-to-day activities, the Test group was found to generally exhibit higher levels of interference. This finding appears somewhat counter intuitive. However, it should be noted that self-reported consumer response data can be subject to bias as a result of the study design with a small sample size. In this study, the hygiene practices of the Test and Control groups were not perfectly matched because the Test group was required to adopt an additional step - the administration mouthwash gargle following brushing. It may be speculated that the addition of this new, adjunctive oral hygiene step may have enhanced the Test group subject's sensitivity to cold/flu symptoms and biased their assessments of symptom interference. Further investigation is necessary to better explain these interference results.

In terms of the methodology employed to derive these results, it must be highlighted that the Wisconsin Upper Respiratory Symptom Survey-21 (WURSS-21) serves as an illness-specific quality of life assessment tool. Its design is geared towards assessing the adverse effects of acute upper respiratory infections, typically presumed to be of viral origin, such as the common cold. This tool has been validated [19] and has been used in recent studies evaluating the effectiveness of glycerol throat spray with cold-adapted cod trypsin [33] in a mouth spray [34] in treating common colds. These studies have reported that the 9-item WURSS-21 domains composite scores served as a sensitive instrument, effectively demonstrating that the treatment significantly enhanced the quality of life for individuals dealing with common colds [33,34]. Additionally, this tool was previously used in studies examining the connection between the severity of acute respiratory infections and inflammatory biomarkers [35]. However, it must be emphasized that WURSSR-21 serves as a specific tool to assess the impact of upper respiratory symptoms in individuals with this condition. As

most of the participants in the present study did not develop any symptom during the 90 days of follow-up, readers must understand that as a possible limitation of the tool. It was noted that symptoms such as a runny nose, nasal congestion, and sneezing showed a stronger association with these biomarkers. It is important to emphasize that, in the current study, participants responded to the WURSS-21 only once at the end of each day over the course of 90 days, providing all self-reported answers.

The present study has limitations that should be acknowledged. Despite the use of a validated tool for assessing common cold-related outcomes, all the results relied on self-reported data. The absence of direct measurements for other clinical cold symptoms conducted by the researchers may introduce a potential gap in generating precise data regarding the manifestation of cold-related symptoms. Despite this limitation, the favorable outcomes associated with the use of CPC+Zn containing mouthwash in reducing the incidence and severity of respiratory symptoms are noteworthy findings. They can provide valuable insights for future research and further contribute to the body of evidence supporting the use of regular cleansing of the posterior oral cavity as a part of daily oral hygiene. Moreover, as only four individuals (2.67% of the whole sample) reported the use of antibiotics during the follow-up period, we believe that the overall results could not change drastically when removing these individuals from the analyses.

## Conclusions

It may be concluded that extending oral care to include the posterior oral cavity through techniques such as gargling with mouthwash has the potential of lowering the incidence of upper respiratory symptoms commonly associated with cold and the flu.

## Supporting information

**S1 Appendix. Full protocol assessed by the ethics committee in Portuguese.**
(DOCX)

**S2 Appendix. Translated version (in English) for the full protocol assessed by the ethics committee.**
(DOCX)

**S3 Appendix. Clinical Trial registration.**
(PDF)

**S4 Appendix. Raw dataset of the study.**
(XLSX)

**S5 Appendix. CONSORT checklist.**
(DOC)

## Acknowledgments

None.

## Author contributions

**Conceptualization:** Francisco Wilker Mustafa Gomes Muniz, Cassiano Kuchenbecker Rösing, Bernal Stewart, Zilson Malheiros, Carlos Benítez, Lyndsay Schaeffer.

**Data curation:** Francisco Wilker Mustafa Gomes Muniz, Maísa Casarin, Natália Marcumini Pola.

**Funding acquisition:** Bernal Stewart, Zilson Malheiros, Carlos Benítez, Lyndsay Schaeffer.

**Investigation:** Francisco Wilker Mustafa Gomes Muniz, Maísa Casarin, Natália Marcumini Pola, Taciane Menezes da Silveira, Francisco Hecktheuer Silva, Guilherme Azario de Holanda, Larissa Viana de Oliveira, Pedro Paulo de Almeida Dantas.

**Methodology:** Francisco Wilker Mustafa Gomes Muniz, Cassiano Kuchenbecker Rösing, Bernal Stewart, Zilson Malheiros, Carlos Benítez, Lyndsay Schaeffer.

**Project administration:** Francisco Wilker Mustafa Gomes Muniz, Bernal Stewart, Zilson Malheiros, Carlos Benítez, Lyndsay Schaeffer.

**Resources:** Bernal Stewart, Zilson Malheiros, Carlos Benítez, Lyndsay Schaeffer.

**Software:** Zilson Malheiros.

**Supervision:** Bernal Stewart, Carlos Benítez.

**Validation:** Maísa Casarin, Natália Marcumini Pola.

**Visualization:** Francisco Wilker Mustafa Gomes Muniz, Taciane Menezes da Silveira, Francisco Hecktheuer Silva, Guilherme Azario de Holanda, Larissa Viana de Oliveira, Pedro Paulo de Almeida Dantas.

**Writing – original draft:** Francisco Wilker Mustafa Gomes Muniz, Maísa Casarin, Natália Marcumini Pola.

**Writing – review & editing:** Cassiano Kuchenbecker Rösing, Taciane Menezes da Silveira, Francisco Hecktheuer Silva, Guilherme Azario de Holanda, Larissa Viana de Oliveira, Pedro Paulo de Almeida Dantas, Bernal Stewart, Zilson Malheiros, Carlos Benítez, Lyndsay Schaeffer.

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
