## [Decision Letter · Decision Letter 0]

16 Oct 2024

PONE-D-24-35124Regular gargling with a Cetylpyridinium Chloride plus Zinc containing mouthwash can reduce upper respiratory symptomsPLOS ONE

Dear Dr. Muniz,

Thank you for submitting your manuscript to PLOS ONE. After careful consideration, we feel that it has merit but does not fully meet PLOS ONE’s publication criteria as it currently stands. Therefore, we invite you to submit a revised version of the manuscript that addresses the points raised during the review process.

I kindly ask you to check the problems raised by the reviewers, so that I can proceed to accept the manuscript.

We look forward to receiving your revised manuscript.

Kind regards,

Maria Giulia Nosotti, Master's Degree

Academic Editor

PLOS ONE

 Journal Requirements: When submitting your revision, we need you to address these additional requirements. 1. Please ensure that your manuscript meets PLOS ONE's style requirements, including those for file naming. The PLOS ONE style templates can be found at https://journals.plos.org/plosone/s/file?id=wjVg/PLOSOne_formatting_sample_main_body.pdf and https://journals.plos.org/plosone/s/file?id=ba62/PLOSOne_formatting_sample_title_authors_affiliations.pdf 2. "Thank you for stating the following financial disclosure: "This study was sponsored by Latin American Oral Health Association and Colgate-Palmolive. The study was also partially funded by Coordenação de Aperfeiçoamento de Pessoal de Nível Superior-Brasil, Finance Code - 001. " Please state what role the funders took in the study.  If the funders had no role, please state: ""The funders had no role in study design, data collection and analysis, decision to publish, or preparation of the manuscript."" If this statement is not correct you must amend it as needed. Please include this amended Role of Funder statement in your cover letter; we will change the online submission form on your behalf. 3. Thank you for stating in your Funding Statement: "This study was sponsored by Latin American Oral Health Association and Colgate-Palmolive. The study was also partially funded by Coordenação de Aperfeiçoamento de Pessoal de Nível Superior-Brasil, Finance Code - 001." Please provide an amended statement that declares *all* the funding or sources of support (whether external or internal to your organization) received during this study, as detailed online in our guide for authors at http://journals.plos.org/plosone/s/submit-now.  Please also include the statement “There was no additional external funding received for this study.” in your updated Funding Statement. Please include your amended Funding Statement within your cover letter. We will change the online submission form on your behalf. 4. Thank you for stating the following in the Competing Interests section: "Drs. Stewart, Malheiros, and Schaeffer are employed by the Colgate-Palmolive Company. Dr. Benítez is employed by the Latin American Oral Health Association." Please confirm that this does not alter your adherence to all PLOS ONE policies on sharing data and materials, by including the following statement: ""This does not alter our adherence to  PLOS ONE policies on sharing data and materials.” (as detailed online in our guide for authors http://journals.plos.org/plosone/s/competing-interests).  If there are restrictions on sharing of data and/or materials, please state these. Please note that we cannot proceed with consideration of your article until this information has been declared.  Please include your updated Competing Interests statement in your cover letter; we will change the online submission form on your behalf. 5. In the online submission form, you indicated that "Data will be available upon request to the corresponding author." All PLOS journals now require all data underlying the findings described in their manuscript to be freely available to other researchers, either 1. In a public repository, 2. Within the manuscript itself, or 3. Uploaded as supplementary information.This policy applies to all data except where public deposition would breach compliance with the protocol approved by your research ethics board. If your data cannot be made publicly available for ethical or legal reasons (e.g., public availability would compromise patient privacy), please explain your reasons on resubmission and your exemption request will be escalated for approval. 6. Please include captions for your Supporting Information files at the end of your manuscript, and update any in-text citations to match accordingly. Please see our Supporting Information guidelines for more information: http://journals.plos.org/plosone/s/supporting-information. 7. Please review your reference list to ensure that it is complete and correct. If you have cited papers that have been retracted, please include the rationale for doing so in the manuscript text, or remove these references and replace them with relevant current references. Any changes to the reference list should be mentioned in the rebuttal letter that accompanies your revised manuscript. If you need to cite a retracted article, indicate the article’s retracted status in the References list and also include a citation and full reference for the retraction notice.

Reviewers' comments:

Reviewer's Responses to Questions

**Comments to the Author**

1. Is the manuscript technically sound, and do the data support the conclusions?

Reviewer #1: Yes

Reviewer #2: Partly

2. Has the statistical analysis been performed appropriately and rigorously? 

Reviewer #1: Yes

Reviewer #2: No

3. Have the authors made all data underlying the findings in their manuscript fully available?

Reviewer #1: Yes

Reviewer #2: Yes

4. Is the manuscript presented in an intelligible fashion and written in standard English?

Reviewer #1: No

Reviewer #2: Yes

5. Review Comments to the Author

Reviewer #1: During the detailed review of this work, we will be able to identify some questions that are not clear and, on the other hand, allow some questions to be asked.

Therefore, I begin my contribution:

1. I thank you once again for trusting me to review this work. Starting with the title, it was not clear what type of analysis was done. If it was a comparison, an efficacy, an effect, an efficiency, I suggest that the title present one of these words that will allow the reader to understand the work much better on a first reading.

2. TITLE: Here in the abstract it is a little clearer. But I suggest that the title be clearer for reading, probably by incorporating the word Effect.

3. INTRODUCTION: In the second paragraph, they begin by talking about the prevalence rates of people who have respiratory infections in the United States of America, since the study originated in Brazil. It would be interesting to present the prevalence and/or incidence rates, so that the reader has a clear idea of the situation in which Brazil finds itself in comparison to the data from the United States.

4. At the end of the introduction. Where the objective of the study is stated, I suggest that it be rewritten to include the word Effect. This will make it clearer and more understandable for the reader.

5. PARTICIPANTS: Within the eligibility criteria. The authors mention that the participants should have good systemic health. What is not clear is whether the patients had any symptoms of respiratory infection. If they did, I suggest that this be written. If not, justify why these patients were not included.

6. Within the exclusion criteria. The authors mention that patients with arthritis and carpal tunnel syndrome could not participate in the study. Why? Justify your answer.

7. EXPERIMENTAL GROUPS: The authors mention the composition of the experimental groups:

a. The negative control group: The authors mention that this group received a toothbrush and a fluoride toothpaste. What is not clear is why this group did not receive a placebo as a product.

8. SAMPLE CALCULATION: It is not clear how the calculation was performed, what it was based on to arrive at the sample number, I suggest that it be explained and rewritten for the readers' understanding. Since this is a phase 3 of a clinical trial

9. RENDERING AND ALLOCATION OF PARTICIPANTS: Upon reading, it is not clear how the participants were allocated to their respective experimental groups, which experimental unit was taken into consideration to allocate these patients. I suggest you specify

10. RESULTS: The authors reported in their results that 01 participant in the control group had adverse effects. One of these patients needed antibiotics for intestinal surgery. What was the justification for remaining in the study? Justify

11. The authors mentioned that 02 participants in the test group used antibiotics. One of them for pain associated with gastritis caused by Helicobacter pillori and the other for presenting a clinical picture of tonsillitis. Why were they not removed from the study? Within the eligibility criteria, it was written by the authors that they should not have used antibiotics. Justify

12. In the results, the authors demonstrated that there was a decrease in hoarseness and chest pressure in the participants in the control group. What is this due to?

13. DISCUSSION: The authors state that due to a subjective test applied to the participants, they demonstrated better results in the absence of upper respiratory symptoms. The questionnaire (WURSSR-21) used serves as a specific tool to assess quality of life in patients with respiratory infection.

Reviewer #2: Major Revision

This appears to be a well conducted clinical trial. However, I am very puzzled by the reporting (see below).

Lines 28 and elsewhere

I suggest replace: ‘Negative Control’ by ‘Control’

Lines 130-132 & 160-161& Tables 1 & 2

Although I am not familiar with the WURSS-21 Daily Symptom Report which, as the authors state, uses the Likert Scale from 0 to 7 but the mean scores given in Tables 1 and 2 are all close to zero with the corresponding SD suggesting negative values are possible. I presume this is a consequence of rescaling to obtain a binary response for analysis. Why not use the Likert Scale itself? This seems very strange so the rational and process used by the authors need to be clearly explained in the paper.

Line 161 95% confidence intervals are mentioned here but are not included in the tables.

Line 163 Since there are only two groups better to use the t-test (however ANOVA is not incorrect)

Lines 164-165

The Tukey multiple comparison test does not appear appropriate when comparing 2 groups.

Tables 1 & 2

Assuming the measures summarised are OK, the difference between the groups needs to be included. For example, Runny nose 0.153 – 0.178 = −0.025 and the 95%CI added.

6. PLOS authors have the option to publish the peer review history of their article (what does this mean? ). If published, this will include your full peer review and any attached files.

**Do you want your identity to be public for this peer review?** For information about this choice, including consent withdrawal, please see our Privacy Policy .

Reviewer #1: No

Reviewer #2: No

---

## [Author Response · Author response to Decision Letter 0]

3 Dec 2024

December 3rd, 2024

TO: Maria Giulia Nosotti

Academic Editor of Plos One.

Dear Dr. Nosotti,

Thank you for the opportunity to review our manuscript ID PONE-D-24-35124 entitled "Efficacy of regular gargling with a Cetylpyridinium Chloride plus Zinc containing mouthwash can reduce upper respiratory symptoms" (new title after peer review) for consideration for future publication in Plos One. We thank the reviewers and the editorial team for their insightful and helpful comments.

We declare that the manuscript is not being considered for publication elsewhere and the results have not been published previously. In addition, we inform that the new final version of the manuscript has been approved by all authors.

As recommended, we have addressed all suggestions punctually and our response to the reviewers can be found on the pages below. All changes were made correctly and are highlighted in the manuscript, with the alterations highlighted in yellow in the document.

Sincerely,

Journal Requirements:

● Answer: Thank you for the opportunity to review and improve our manuscript. We have updated the manuscript to follow the Journal's style requirements. Both templates were considered in the new version of the manuscript.

2. "Thank you for stating the following financial disclosure:

"This study was sponsored by the Latin American Oral Health Association and Colgate-Palmolive. The study was also partially funded by Coordenação de Aperfeiçoamento de Pessoal de Nível Superior-Brasil, Finance Code - 001. "

Please state what role the funders took in the study. If the funders had no role, please state: "The funders had no role in study design, data collection and analysis, decision to publish, or preparation of the manuscript.""

● Answer: The cover letter was updated to clearly demonstrate the role of funders in the present study. The following sentence was included: "Role of funders: The funders shared a role with the other authors in study design, data analysis, and preparation of the manuscript. However, funders had no direct role in data collection, they only supervised data collection."

"This study was sponsored by the Latin American Oral Health Association and Colgate-Palmolive. The study was also partially funded by Coordenação de Aperfeiçoamento de Pessoal de Nível Superior-Brasil, Finance Code - 001."

Please provide an amended statement that declares *all* the funding or sources of support (whether external or internal to your organization) received during this study, as detailed online in our guide for authors at http://journals.plos.org/plosone/s/submit-now. Please also include the statement "There was no additional external funding received for this study." in your updated Funding Statement.

● Answer: Funding Statement was included in the cover letter with the following sentences: "Funding Statement: This study was sponsored by the Latin American Oral Health Association and Colgate-Palmolive. The study was also partially funded by Coordenação de Aperfeiçoamento de Pessoal de Nível Superior-Brasil, Finance Code - 001. There was no additional external funding received for this study."

"Drs. Stewart, Malheiros, and Schaeffer are employed by the Colgate-Palmolive Company. Dr. Benítez is employed by the Latin American Oral Health Association."

Please confirm that this does not alter your adherence to all PLOS ONE policies on sharing data and materials, by including the following statement: ""This does not alter our adherence to PLOS ONE policies on sharing data and materials." (as detailed online in our guide for authors http://journals.plos.org/plosone/s/competing-interests). If there are restrictions on sharing of data and/or materials, please state these. Please note that we cannot proceed with consideration of your article until this information has been declared.

● Answer: This information was included in the cover letter with the following sentences: "Competing interest statement: Drs. Stewart, Malheiros, and Schaeffer are employed by Colgate-Palmolive Company. Dr. Benítez is employed by the Latin American Oral Health Association. This does not alter our adherence to PLOS ONE policies on sharing data and materials."

5. In the online submission form, you indicated that "Data will be available upon request to the corresponding author."

● Answer: The authors confirm that the data supporting the findings of this study are available within the article. The raw data was uploaded in the Journal's system, which is also disclosed in the main text. The following sentence stated it: "The raw data is available in the Appendix 1."

● Answer: All supporting information are cited in the main text with the following sentences:

o The study protocol had previously received approval from the Ethics Committee of the School of Dentistry at the Federal University of Pelotas, under protocol CAAE 58924622.7.0000.5318 (Appendixes 1-4).

o The study protocol was registered a posteriori in Clinical Trials database (NCT06479226) (Appendix 5).

o The raw data is available in the Appendix 6.

o The study followed the CONSORT checklist (Appendix 7).

At the end of the manuscript, the captions were provided, which now reads:

Captions for the Supporting Information files

Appendix 1. Approval from the local ethics committee in Portuguese.

Appendix 2. Translated version (in English) for the approval from the local ethics committee.

Appendix 3. Full protocol assessed by the ethics committee in Portuguese.

Appendix 4. Translated version (in English) for the full protocol assessed by the ethics committee.

Appendix 5. Clinical Trial registration.

Appendix 6. Raw dataset of the study.

Appendix 7. CONSORT checklist.

● Answer: The whole list of references follows the Journal's style. Moreover, we have checked if any cited study was retracted. That was not the case for all of them.

Reviewer #1: During the detailed review of this work, we will be able to identify some questions that are not clear and, on the other hand, allow some questions to be asked.

● Answer: Thank you very much for the effort in reviewing and improving our manuscript. The authors are available for further questions.

Therefore, I begin my contribution:

1. I thank you once again for trusting me to review this work. Starting with the title, it was not clear what type of analysis was done. If it was a comparison, an efficacy, an effect, an efficiency, I suggest that the title present one of these words that will allow the reader to understand the work much better on a first reading.

● Answer: We understand that this is a study of efficacy that compared two experimental protocols for an outcome of upper respiratory symptoms. Therefore, the title was changed to: "Efficacy of regular gargling with a Cetylpyridinium Chloride plus Zinc containing mouthwash can reduce upper respiratory symptoms."

2. TITLE: Here in the abstract it is a little clearer. But I suggest that the title be clearer for reading, probably by incorporating the word Effect.

● Answer: The title was changed to: "Efficacy of regular gargling with a Cetylpyridinium Chloride plus Zinc containing mouthwash can reduce upper respiratory symptoms."

3. INTRODUCTION: In the second paragraph, they begin by talking about the prevalence rates of people who have respiratory infections in the United States of America, since the study originated in Brazil. It would be interesting to present the prevalence and/or incidence rates, so that the reader has a clear idea of the situation in which Brazil finds itself in comparison to the data from the United States.

● Answer: The following sentences were included to clearly demonstrate the prevalence of respiratory symptoms in Brazil, including its seasonality based on the weather: "A population-based cross-sectional study conducted in Brazil in May 2020 reported a prevalence of flu-like syndrome symptoms of 3.38% [7]. Furthermore, influenza incidence in Brazil is influenced by seasonal variation, with higher rates observed in winter and the lowest rates recorded in January during the summer months [8]."

4. At the end of the introduction. Where the objective of the study is stated, I suggest that it be rewritten to include the word Effect. This will make it clearer and more understandable for the reader.

● Answer: As previously stated, we believed that this is an efficacy study. Therefore, the mentioned sentence now reads: "The present study assessed the efficacy of regular gargling with a mouthwash containing CPC+Zn in reducing the incidence of upper respiratory symptoms."

5. PARTICIPANTS: Within the eligibility criteria. The authors mention that the participants should have good systemic health. What is not clear is whether the patients had any symptoms of respiratory infection. If they did, I suggest that this be written. If not, justify why these patients were not included.

● Answer: The present study did not aim to treat upper respiratory symptoms with the proposed experimental groups. Therefore, individuals with any symptoms during the baseline appointment were not included, as it was aimed to detect the reduction in the incidence of those symptoms. This is now reported under the following sentences: "2) Be in good general health as determined by the study investigators, which included the absence of any respiratory symptoms at baseline. Participants with upper respiratory symptoms were not included, as the current study did not aim to treat these conditions;"

6. Within the exclusion criteria. The authors mention that patients with arthritis and carpal tunnel syndrome could not participate in the study. Why? Justify your answer.

● Answer: Literature reports that individuals with arthritis might have poorer oral hygiene behaviors, including a higher prevalence of oral diseases (Chang, Y., Chung, M. K., Park, J. H., & Song, T. J. (2023). Association of oral health with risk of rheumatoid arthritis: a nationwide cohort study. Journal of Personalized Medicine, 13(2), 340.). Moreover, it was hypothesized that those with carpal tunnel syndrome would perform their oral hygiene with a lower efficiency when compared to those without it. Based on this information, to homogeneity the sample, decreasing potential biases, these individuals were not included in the present study.

This information was reported under the following sentences: "9) Carpal tunnel or arthritis in their hands. These individuals were not included as literature shows poorer oral hygiene behaviors, including higher prevalence of oral diseases, among those with arthritis [18]. Moreover, it was hypothesized that those with carpal tunnel syndrome would perform their oral hygiene in a lower efficiency when compared to those without it."

7. EXPERIMENTAL GROUPS: The authors mention the composition of the experimental groups:

a. The negative control group: The authors mention that this group received a toothbrush and a fluoride toothpaste. What is not clear is why this group did not receive a placebo as a product.

● Answer: We acknowledge this limitation. However, when preparing the protocol for the present study, we aimed to compare individuals who perform the cleansing of the throat when compared to those without it. To clearly state that, the following sentence was included in this section: "No placebo substance was used in the control group, as the current study aimed to compare individuals who perform the cleansing of their throat with CPC+Zn in comparison to those who did not perform it."

8. SAMPLE CALCULATION: It is not clear how the calculation was performed, what it was based on to arrive at the sample number, I suggest that it be explained and rewritten for the readers' understanding. Since this is a phase 3 of a clinical trial

● Answer: The study was powered to detect a difference of 0.03 in the proportion of subjects exhibiting at least one symptom. This calculation assumed a within group standard deviation of 0.53, 80% power, and approximately 75 daily observations per participant. The sentence was rewritten to: "A power calculation indicated that with approximately 75 observations for each participant, the study can detect a difference in incidence rates of 3% with 80% probability."

9. RENDERING AND ALLOCATION OF PARTICIPANTS: Upon reading, it is not clear how the participants were allocated to their respective experimental groups, which experimental unit was taken into consideration to allocate these patients. I suggest you specify

● Answer: A list of randomization was performed by a researcher not involved in the clinical examination. This procedure was performed using a website specifically developed for it (https://www.randomization.org). Two experimental groups were specified in the website, and individuals were allocated in the order determined by the website and based on the order of their inclusion. This information is provided by the following sentences: "The randomization process was performed by a researcher (NMP) who did not take part in the clinical examinations. Simple randomization was employed, and a website (https://www.randomization.org) was utilized for this purpose. This process was performed in blocks of different sizes."

10. RESULTS: The authors reported in their results that 01 participant in the control group had adverse effects. One of these patients needed antibiotics for intestinal surgery. What was the justification for remaining in the study? Justify

● Answer: At baseline, as previously stated, all individuals should be in good general health. This would be the only way to clearly show the impact of the proposed intervention on the new events of upper respiratory symptoms. Therefore, those using antibiotics or with recent use of antibioti

---

## [Decision Letter · Decision Letter 1]

17 Dec 2024

Efficacy of regular gargling with a Cetylpyridinium Chloride plus Zinc containing mouthwash can reduce upper respiratory symptoms

PONE-D-24-35124R1

Dear Dr. Muniz,

We’re pleased to inform you that your manuscript has been judged scientifically suitable for publication and will be formally accepted for publication once it meets all outstanding technical requirements.

Kind regards,

Maria Giulia Nosotti, Master's Degree

Academic Editor

PLOS ONE

Additional Editor Comments (optional):

Reviewers' comments:

Reviewer's Responses to Questions

**Comments to the Author**

1. If the authors have adequately addressed your comments raised in a previous round of review and you feel that this manuscript is now acceptable for publication, you may indicate that here to bypass the “Comments to the Author” section, enter your conflict of interest statement in the “Confidential to Editor” section, and submit your "Accept" recommendation.

Reviewer #2: All comments have been addressed

2. Is the manuscript technically sound, and do the data support the conclusions?

Reviewer #2: Yes

3. Has the statistical analysis been performed appropriately and rigorously? 

Reviewer #2: Yes

4. Have the authors made all data underlying the findings in their manuscript fully available?

Reviewer #2: Yes

5. Is the manuscript presented in an intelligible fashion and written in standard English?

Reviewer #2: Yes

6. Review Comments to the Author

Reviewer #2: The authors have responded appropriately to suggestions made in my earlier review I have no further comments.

7. PLOS authors have the option to publish the peer review history of their article (what does this mean? ). If published, this will include your full peer review and any attached files.

**Do you want your identity to be public for this peer review?** For information about this choice, including consent withdrawal, please see our Privacy Policy .

Reviewer #2: No

---

## [Editor Report · Acceptance letter]

PONE-D-24-35124R1

PLOS ONE

Dear Dr. Muniz,

I'm pleased to inform you that your manuscript has been deemed suitable for publication in PLOS ONE. Congratulations! Your manuscript is now being handed over to our production team.

Kind regards,

on behalf of

Dr. Maria Giulia Nosotti

Academic Editor

PLOS ONE